# Ceruloplasmin and Coronary Heart Disease—A Systematic Review

**DOI:** 10.3390/nu12103219

**Published:** 2020-10-21

**Authors:** Antonio P. Arenas de Larriva, Laura Limia-Pérez, Juan F. Alcalá-Díaz, Alvaro Alonso, José López-Miranda, Javier Delgado-Lista

**Affiliations:** 1Lipids and Atherosclerosis Unit, Department of Internal Medicine, Maimonides Biomedical Research Institute of Cordoba (IMIBIC), Reina Sofia University Hospital, University of Cordoba, Av. Menendez Pidal s/n, 14004 Cordoba, Spain; h52arlaa@uco.es (A.P.A.d.L.); lauralimiaperez@gmail.com (L.L.-P.); jlopezmir@uco.es (J.L.-M.); md1delij@uco.es (J.D.-L.); 2CIBER Fisiopatología de la Obesidad y Nutrición (CIBEROBN), Instituto de Salud Carlos III (ISCIII), 28029 Madrid, Spain; 3Department of Epidemiology, Rollins School of Public Health, Emory University, Atlanta, GA 30322, USA; alvaro.alonso@emory.edu

**Keywords:** ceruloplasmin, coronary heart disease, inflammation

## Abstract

Several studies indicate that oxidative stress might play a central role in the initiation and maintenance of cardiovascular diseases. It remains unclear whether ceruloplasmin acts as a passive marker of inflammation or as a causal mediator. To better understand the impact of ceruloplasmin blood levels on the risk of cardiovascular disease, and paying special attention to coronary heart disease, we conducted a search on the two most commonly used electronic databases (Medline via PubMed and EMBASE) to analyze current assessment using observational studies in the general adult population. Each study was quality rated using criteria developed by the US Preventive Services Task Force. Most of 18 eligible studies reviewed support a direct relationship between ceruloplasmin elevated levels and incidence of coronary heart disease. Our results highlight the importance of promoting clinical trials that determine the functions of ceruloplasmin as a mediator in the development of coronary heart disease and evaluate whether the treatment of elevated ceruloplasmin levels has a role in the prognosis or prevention of this condition.

## 1. Introduction

Cardiovascular disease (CVD) is the largest cause of death worldwide in developed countries. As a diagnostic category, CVD includes various areas: coronary heart disease (CHD), manifested by myocardial infarction (MI) or angina pectoris; cerebrovascular disease, manifested by stroke and transient ischemic attack; high blood pressure; peripheral artery disease and death by any of the above causes [1].

Despite the fact that the mortality from CHD has decreased over the last few decades in western countries, it still causes about one-third of all deaths in people over 35 years [2,3,4]. Although current guideline-guided CHD therapy has lowered both recurrence and death rates, people with CHD remain at high risk for these complications. One third of all CHD with known, controlled risk factors will have a recurrence in the following 10 years [5]. This is called residual risk, and many approaches have been taken to tackle it. One of the most important fields of research in this area is the search for additional biomarkers which may help to detect or be an early predictor of those cases which will develop a worse prognosis despite controlled risk factors.

Inflammation and oxidative stress are two of the processes involved in the development of atherosclerosis and CHD. Oxidative stress is believed to be a consequence of increased circulating neurohormones and hemodynamic disorder. Impairment of cardiac function could be caused by a redox balance disorder, an oxidative damage to cellular molecules, or a damage in cell signaling, compromising the cell survival and leading to death [6,7].

Ceruloplasmin (CP) belongs to the α2-glycoprotein fraction of plasma proteins. It is synthesized in the liver, incorporating copper, mainly from the diet, and accounts for 95% of the total circulating copper in healthy adults. Apart from playing a role in copper and iron metabolism, CP is an acute-phase reactant that may work as an antioxidant but can also generate free radicals that may lead to several illnesses [8,9]. It is interesting that most of the plasma copper that will end up building CP comes from dietary copper consumed weeks or months ago, not from recent meals, so it will take some time for CP to reflect changes involving copper availability in the diet.

Oxidative stress might play a central role in the initiation of CVD, but it remains unclear whether CP acts as a passive marker of inflammation or as a causal mediator in its development. Reviewing the scientific literature, it is clear that elucidating the effect of CP on CHD is a difficult task. For this reason, we carried out the first systematic review which provides evidence from the observational studies involving the effect of CP over the last three decades, in an attempt to better understand its impact on the risk of CHD.

## 2. Methods

We conducted a search of the two most commonly used electronic databases (Medline via PubMed and EMBASE) to analyze current assessment in observational studies. We identified the records using the following keywords: “Cardiovascular” OR “coronary” OR “heart” OR “angina” OR “myocardial” OR “infarction” AND “ceruloplasmin.” Equivalent free-text terms were used. In this review, we were interested in exploring the evidence of CP and CHD after 1990.

The search resulted in 1407 records which were categorized and screened independently by AAL and LLP (differences resolved by JDL). When analyzing original articles, the authors decided whether the item was relevant or not, based on the title and the abstract. If considered pertinent, the referenced articles included in the item were added to the list of potential articles to include in this review. To assess the validity of each of these studies, we reviewed all the related articles and evaluated the quality of each study on the basis of criteria created by the third USPSTF (US Preventive Services Task Force) (Table 1) [10].

We removed all articles written in languages other than English, duplicated records, reviews, conference abstracts and letters, studies on a pediatric population, studies on animals and unrelated articles, resulting in 18 eligible studies (Table 2). This is illustrated as a flow-chart according to the PRISMA statement in Figure 1.

## 3. Results

The probable relationship between the CP concentration in serum and the incidence of atherosclerosis and other cardiovascular conditions was first suggested in the 1950s [30]. High serum CP levels have been found in patients with arteriosclerosis [31], unstable angina [32], stroke and MI (15). Prospective studies have showed that serum CP may be an independent risk factor for cardiovascular disease. A nested, case-control study by Reunanen et al. was the first to show that serum CP was positively associated with CHD [12]. In this study, which included 208 patients, the association between serum CP level and the subsequent incidence of MI was explored and higher serum CP concentration was found to be directly related with MI with an adjusted Odds Ratio (OR) of 3.1 (95% confidence interval (CI) 1.3–7.6) in the highest tertile compared to the lowest tertile. The same result was obtained by Mänttäri et al. in middle-aged patients with dyslipidemia, where there was a continuous, graded increase in coronary risk in patients with increasing CP. The risk in the highest tertile was double (OR 2.1; 95% CI 1.1–4.2) that of the lowest tertile, with an odds ratio of 2.4 (95% CI 1.3–4.4) in subjects with high low-density lipoprotein cholesterol and of 11.3 (95%CI 2.5–52.2) in subjects with low high-density lipoprotein cholesterol [13].

Because inflammation is recognized as a key player in atherosclerotic progression, Mori et al. separated the risk contributed by CP from that of inflammation (α1-antitrypsin, α1-acid glycoprotein, α2-macroglobulin, haptoglobin, fibrinogen, C4b binding protein, lipoprotein (a) and C-reactive protein (CRP)) and suggested that CP could serve as independent risk factor for coronary atherosclerosis and as a marker for the severity of disease [14]. In these terms, Klipstein-Grobusch et al. confirmed the association between serum CP levels and subsequent MI in a four-year follow-up study. The risk of MI for the highest compared with the lowest quartile of CP was 2.46 (95% CI 1.04–6.00), a relationship that continued after making adjustments for other potential contributors, like C-reactive protein or leucocyte count (lowering a third of the effect), thus supporting the theory that the excess CP itself contributes substantially to the risk of CHD [16].

Five inflammation-sensitive plasma proteins (ISPs; fibrinogen, oroso-mucoid, α1-antitrypsin, haptoglobin, and CP) were measured in 6075 healthy patients from the Malmö Study with a mean follow-up of 18 years. Engstrom et al. carried out several studies in this population to determine the association of ISPs with incidence of CHD, the cardiovascular risk in overweight or obese men and the fatality of future coronary events.

In the first study, the incidence of MI was related to ISPs. Patients were categorized into low-risk and high-risk groups according to traditional risk factors. The relative risk in the highest quartile of low-risk group were 1.00 (reference), 1.9 (95% CI 0.8–4.2), 1.8 (95% CI 0.6–5.4), and 2.9 (95% CI 1.05–8.1), respectively, for men with an increasing number of ISPs (0, 1, 2 and ≥3 ISPs). On the other hand, in the high-risk group, RRs were 1.00, 1.4 (95% CI 0.9–2.2), 1.9 (95% CI 1.2–3.1), and 2.0 (95% CI 1.3–3.1), respectively, for men with an increasing number of ISPs (0, 1, 2 and ≥3 ISPs) [17].

The idea that ISPs may modify the cardiovascular risk in overweight or obese men was explored in a later study. High ISPs levels were associated with an increased cardiovascular risk. The age-adjusted relative risks in obese men were 2.1 (95% CI 1.4–3.4), 2.4 (95% CI 1.5–3.7), 3.7 (95% CI 2.3–6.0), and 4.5 (95% CI 3.0–6.6), respectively, for men with an increasing number of ISPs (0, 1, 2 and ≥3 ISPs) [18].

A third study was performed to determine whether low-grade inflammation (measured by ISP levels) in healthy men predicted the fatality of future coronary events. A higher number of CHD deaths was noted in men who had presented a low-grade inflammation during many previous years. Of the 680 men with a coronary event, 197 died on the first day and 228 died within 28 days. The proportions who died on the first day were 26%, 25%, 29%, and 35%, respectively, for men with an increasing number of ISPs (0, 1, 2 and ≥3 ISPs). The corresponding proportions who died within 28 days were 30%, 31%, 34%, and 38%, respectively. Although we cannot infer the exact contribution of CP to these studies, it seems that it may interact with some of the other inflammatory markers, and might exert its hypothetical effect by means of inflammation [19].

Systolic dysfunction in acute ST-elevation MI patients seems to be associated with an inflammatory response characterized by a rise in plasma concentration of ISPs (α1antitrypsin, α1glycoprotein, haptoglobin, CP and C-reactive protein) and appears to provide a short-term prognostic relevance. This was the conclusion reached by Brunetti et al., where the incidence of CHD correlated with the ISP values, and CP values were the most significant markers of acute heart failure when compared with patients without systolic dysfunction (40.1 ± 9.7 vs. 31.4 ± 7.6 mg/dL, *p* < 0.001) [21].

Göçmen et al. found increased CP levels in CHD patients. An increase in oxidants could be a possible cause of this increase in CP in CHD patients. In their study, serum CP levels were reported to be an independent risk factor for cardiovascular diseases [22]. There was a consistent increase in the levels of CP in CHD patients (48.93 ± 4.44 mg %) compared to controls (32.25 ± 4.67 mg %) in a study by Kaur et al., where statistical analysis revealed significantly increased CP levels in all the subgroups (acute MI, unstable angina and stable angina) [23].

CP levels were elevated in patients with acute MI and Diabetes Mellitus (DM) compared to non-diabetics with MI, possibly because of the greater degree of inflammation in these patients. However, regardless of this factor, CHD patients showed more inflammation, and consequently higher CP levels, than controls [24].

In a prospective case-control study by Kumar et al., designed to identify and evaluate potential risk factors in normolipidemic patients with an acute MI, CP was again found to be higher in cases than in the controls, in a study that also evaluated other potential markers, such as lipoprotein a, CRP and fibrinogen [25].

Subclinical myocardial necrosis as a prognostic feature of long-term adverse cardiac events risk has been studied on several occasions. Tang et al. first explored it in 3828 patients undergoing elective diagnostic coronary angiography with troponin I levels below the cut-off for defining MI. Here, the authors studied the underlying mechanisms contributing to myocardial necrosis and the risk of major adverse cardiovascular events. CRP and CP were associated with subclinical myocardial necrosis after 3-years of follow-up [26]. Afterwards, Tang et al. studied the relationship between CP levels and the incidence of major adverse cardiovascular events (MACE = death, MI and stroke) in 4177 patients undergoing elective coronary angiography after 3-years of follow-up. They performed a genome-wide association study of CP to determine genetic variants that could be related to prevalent and incident cardiovascular risk. Serum CP level was associated with higher risk of MI with a HR of 2.35, (95% CI 1.79–3.09) when comparing the top quartile versus the lowest. CP remained independently predictive of MACE (HR 1.55, 95% CI 1.10–2.17). Genetic variants at the CP locus were not associated with prevalent or incident risk of CAD [27].

Similar conclusions were reached by Grammer et al., who examined whether serum copper and CP concentrations were associated with angiographic CAD and mortality from all causes and cardiovascular causes. When the highest quartile for CP levels was compared to the lowest, HR for death from any cause was 2.63 (95% CI, 2.17–3.20), and HR for death from cardiovascular causes was 3.02 (95% CI, 2.36–3.86). The concentration of CP was therefore independently associated with increased risk of death from all and cardiovascular causes [28].

Finally, in a sub-cohort of 4658 participants in the Malmö Diet and Cancer study, seven inflammatory markers were studied to evaluate their incidence in DM and CVD (coronary events, including fatal and nonfatal MI or stroke). CP, among other molecules, predicted an increased risk of CVD but not of DM [29].

However, not all the studies evaluating the relationship between serum CP and CHD have produced the same results. For example, Enbergs et al. found no such relation when attempting to relate CP with the extent of atherosclerosis in coronarography. However, there were a number of limitations as regards patient selection in this investigation [15].

Along similar lines, Verma et al. explored how serum levels of three antioxidants (vitamin C, bilirubin and CP) were related to CHD risk factors. A 7–18% decrease was observed in CHD patients with severe disease, with increasing serum levels of the three antioxidants. Similarly, a decrease of 14–20% was objectified in triple vessel disease and of 39% in MI with increasing serum CP in CHD patients, compared to the non-MI group. An inverse relationship was found between the three antioxidants studied and coronary risk factor suggesting that greater care in traditional risk factors would maintain a high level of these antioxidants [20].

## 4. Discussion

Our systematic review shows an association between elevated CP levels and CHD that is unrelated to Framingham risk factors. As mentioned above, traditional risk factors are thought to account for most CHDs, although 15% to 20% of patients have no identified risk factors and this entails the impossibility of an adequate treatment to prevent a first event. For this reason, the scientific community has tried to identify other modifiable risk factors which help to predict an important number of CHD events. For this, CP, a protein closely linked to inorganic nutrient copper, is a strong, biologically-plausible candidate.

Most of the studies analyzed in the present work showed a uniformity in their results, with the exception of the study by Verma et al., towards a direct relationship between serum CP levels and incidence of CHD or cardiovascular events. The higher serum CP level the patient has, the more likely the patient is to suffer some complications.

The mechanism by which CP may influence cardiovascular disease is still unknown. It seems that reactive oxygen species (ROS), such as superoxide and hydrogen peroxide, may be important in the main underlying mechanisms. It is hypothesized that, with high ROS levels, antioxidant systems such as superoxide dismutase, catalase and glutathione are overwhelmed and the structural integrity of CP is damaged [33].

CP is involved in iron removal from the cells and its dysfunction, in particular a loss of its ferroxidase activity, may lead to an accumulation of iron in tissues. Moreover, this loss of function produces unbound or free copper, which, together with the iron, can produce pathogenic effects in the cell such as apoptosis, cell toxicity, cell replication, greater oxidative stress and pathogenic gene activation [9]. It has been shown in a murine model that copper from myocardium is also released to the blood during the ischemic process thanks to the increase of copper metabolism MURR domain 1 (COMMD1), a Cu transport chaperone [34]. Therefore, the increase of copper in blood, and secondarily of CP caused by MI, seems not completely associated with an inflammatory response.

Oxidation of Low-density lipoproteins (LDLs) leads to the initiation or acceleration of the atherosclerotic process inducing the formation of autoantibodies against oxidized LDL (anti-oxLDL). The presence of CP and other acute-phase proteins in atherosclerotic lesions seems to incriminate a pathway involving lipid and lipoprotein oxidation, which plays an important role in the etiology of CHD. In fact, several study groups have provided evidence that CP is a potent catalyst of LDL oxidation in vitro and in vivo [35,36,37]. The study published by Awadallah et al. deserves particular consideration being the only study demonstrating an association between the concentrations of anti-oxLDL and those of CP and copper in patients with CVD [38]. Serum concentrations of CP and LDL lipid peroxides were correlated with atherosclerotic process and restenosis in patients undergoing endarterectomy [39]. Therefore, these studies suggest that CP may play a role in the oxidation of LDL in vivo.

To summarize and tie all the above threads together, CP may promote an inflammatory environment, with the associated defense mechanisms activating the ROS cascade by directly or indirectly producing the oxidation of LDL.

Another important mechanism by which CP may exert an effect in atherosclerosis is by affecting the nitric oxide (NO) pathway, which plays a key role in normal cardiac physiology and a protective role in the ischemic and failing heart [40,41]. CP can exert an important prooxidant function, related to NO oxidase, which may decrease NO bioavailability in plasma through its catalytic activity under given conditions. This has been shown in several studies after CP immuno-depletion and in humans with aceruloplasminemia with no NO oxidase activity [42,43].

Although most of the studies support a direct association between serum CP and CHD, some studies have questioned this association. In this context, CP was shown by Chapman et al. to have another antioxidant property through inhibition of myeloperoxidase, which stops free radical production [44], and Verma et al. established an inverse relationship between serum CP levels and coronary risk factors [20]. However, this study also had several limitations: CP was measured with an old technique and the international system of units was not followed; the baseline characteristics of the patients were not defined and only cholesterol and triglycerides were compared as CHD risk factors with the three antioxidants. Finally, Enbergs et al. did not find any quantitative relationship between CP and the coronary atherosclerosis observed through angiography. Nevertheless, the paper also had one very critical limitation, in that it excluded patients with inflammatory conditions, thus consequently ruling out a clinical population with increased CP levels.

## 5. Limitations

As in any review, publication bias may have shifted the review towards positive findings, due to the fact that positive results have more chances of being published. Nevertheless, the fact that a good proportion of our results come from studies where CP is not the main objective, and results of those article are aligned with the articles in which CP is the main focus, supports our results. Another limitation is that we were not able to establish a specific cut-off point of CP to indicate a higher CHD risk. This is because studies are not homogeneous and so neither are their results. The limits established in our search strategy may also have limited our results. Although some articles written in other languages may be valid, limiting the articles included to those written in English is a common feature in Reviews. Finally, we reviewed here the articles published in the last 30 years, but not earlier studies.

## 6. Conclusions

Most of the studies reviewed in this article support a direct relationship between elevated CP levels and CVD. Patients with high CP serum levels were more likely to have a CV event, especially a CHD. However, CP cannot currently be considered as a coronary risk factor that may provide any value in prioritizing preventive interventions amongst those with unrecognized CVD or offer recommendations for people in secondary CHD prevention.

In view of the results of the observational studies included in this review, we believe that there is a basis supporting the importance of evaluating whether the treatment of elevated CP levels has a role in the prognosis or prevention of CHD.

## Figures and Tables

**Figure 1 nutrients-12-03219-f001:**
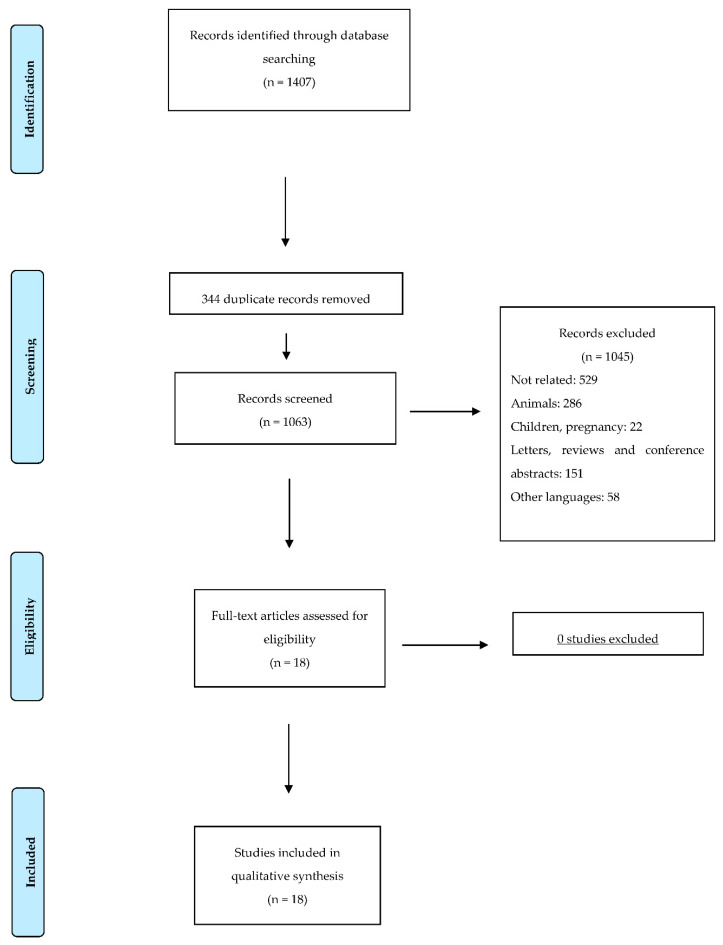
Flow diagram illustrating the search strategy used according to PRISMA (Preferred Reporting Items for Systematic reviews and Meta-Analyses) statement.

**Table 1 nutrients-12-03219-t001:** US Preventive Services Task Force Quality Rating Criteria *.

**Cohort studies** Criteria Initial assembly of comparable groups: cohort studies—consideration of potential confounders with either restriction or measurement for adjustment in the analysis; consideration of inception cohorts Maintenance of comparable groups (including attrition, crossovers, adherence, contamination) Major differential loss in follow-up or overall high loss in follow-up Measurements: equal, reliable and valid (including masking of outcome assessment) Clear definition of interventions Important outcomes considered Definition of ratings on the basis of above criteria **Good** Meets all criteria: comparable groups are assembled initially and maintained throughout the study (follow-up at least 80%); reliable and valid measurement instruments are used and applied equally to the groups; important outcomes are considered, and appropriate attention is given to confounders in analysis. **Fair** Any or all of the following problems occur, without the important limitations noted in the “poor” category: generally comparable groups are assembled initially but some question remains about whether some (albeit not major) differences occurred in follow-up; measurement instruments are acceptable (albeit not the best) and are generally applied equally; some but not all important outcomes are considered, and some but not all potential confounders are accounted for. **Poor** Any of the following major limitations exists: groups assembled initially are not close to being comparable or maintained throughout the study; unreliable or invalid measurement instruments are used, and key confounders are given little or no attention.
**Case-control studies** Criteria Accurate ascertainment of cases Non-biased selection of cases/controls with exclusion criteria applied equally to both Response rate Diagnostic testing procedures applied equally to each group Measurement of exposure accurate and applied equally to each group Appropriate attention to potential confounding variables Definition of ratings on the basis of the above criteria **Good** Appropriate ascertainment of cases and non-biased selection of case and control participants; exclusion criteria applied equally to cases and controls; response rate equal to or greater than 80%; diagnostic procedures and measurements accurate and applied equally to cases and controls, and appropriate attention to confounding variables. **Fair** Recent, relevant, without major apparent selection or diagnostic work-up bias but with response rate less than 80% or attention to some but not all important confounding variables. **Poor** Major selection or diagnostic work-up biases, response rates less than 50%, or inattention to confounding variables.

* Adapted from Humphrey et al. [11].

**Table 2 nutrients-12-03219-t002:** Summary of the articles included in this review.

*Authors and Year of Publication*	*Study Design, Population; Age (y; Mean ± Standard Desviation)*	*Sample Size, Cases/Controls*	*Follow-Up in Years (If Applicable)*	*Outcomes Evaluated*	*Main Findings Related to Ceruloplasmin*	*Quality of Study*	*Supports a Direct Relationship between Higher Ceruloplasmin (CP) Levels and Coronary Heart Disease (CHD) Risk (Yes/No)*
***Reunanen et al.*** [12]***; 1992***	Nested case-control, men and women; 59 (mean)	104/104	11.0	Incidence of MI and stroke	Higher serum CP levels are a risk factor for myocardial infarction (MI). Adjusted OR in the highest tertile: 3.1 (1.3–7.6 95% confidence interval (CI)	Good	Yes
***M. Manttari et al.*** [13]***; 1994***	Nested case-control, men; 49.3 ± 4.4 (cases); 47.2 ± 4.7 (controls)	136/136	5.0	Non-fatal myocardial infarction or cardiac death	There was an increase in coronary risk in patients with rising CP. The risk in the highest tertile was double (OR 2.1; 1.1–4.2 95% CI) that of the lowest. The risk of high CP was influenced by lipoprotein cholesterol concentrations, with an odds ratio of 2.4 (1.3–4.4 95% CI) in subjects with high low-density lipoprotein cholesterol and of 11.3 (2.5–52.2 95% CI) in subjects with low high-density lipoprotein cholesterol.	Good	Yes
***Mori et al.*** [14]***; 1995***	Cohort, men and women, 57.8 ± 9.7; 61.2 ± 9.3 respectively	225	4.1	Severity of coronary atherosclerosis in patients undergoing coronary angiography. (Gensini Score)	CP can be an independent risk factor for coronary atherosclerosis and determine the severity of the disease.	Fair	Yes
***Enbergs et al.*** [15]***; 1998***	Cohort, men and women, 55.1 ± 9.6; 54.6 ± 10.0 respectively.	275	1.0	The extent of CHD assessed by three scores (Vessel score, stenosis score and extent score)	Serum CP levels were not confirmed as risk factor for the extent of CHD.	Fair	No
***Klipstein-Grobusch et al.*** [16]***; 1999***	Nested case-control, men and women; 76.4 ± 8.7 (cases); 76.8 ± 9.0 (controls)	83/127	4.0	Incidence of MI	Risk of MI for the highest compared with the lowest quartile of CP was 2.46 (1.04–6.00 95% CI). After adjustment for C-reactive protein and leucocyte count, the excess risk was reduced by 33% suggesting that the association between serum CP and CHD may be attributed to inflammation processes.	Good	Yes
***G. Engström et al.*** [17]***; 2003***	Cohort, men; mean approximately 46.9	6075	18.1 ± 4.3 years	Incidence of MI	CP levels increased the Incidence of MI. The relative risk in the highest quartile of low-risk group were 1.00 (reference), 1.9 (95% CI 0.8–4.2), 1.8 (95% CI 0.6–5.4), and 2.9 (95% CI 1.05–8.1), respectively, for men with an increasing number of inflammation-sensitive plasma proteins (ISPs) (0, 1, 2 and ≥ 3 ISPs). On the other hand, in the high-risk group, relative risks (RRs) were 1.00, 1.4 (95% CI 0.9–2.2), 1.9 (95% CI 1.2–3.1), and 2.0 (95% CI 1.3–3.1), respectively, for men with an increasing number of ISPs (0, 1, 2 and ≥ 3 ISPs)	Good	Yes
***G. Engström et al.*** [18]***; 2004***	Cohort, men; 46.8 ± 3.7.	6075	18.7 ± 4.2	Incidences of cardiovascular events (myocardial infarction, stroke, cardiovascular deaths), cardiac events (fatal or nonfatal myocardial infarction), and stroke	The age-adjusted relative risks in obese men were 2.1 (95% CI 1.4–3.4), 2.4 (95% CI 1.5–3.7), 3.7 (95% CI 2.3–6.0), and 4.5 (95% CI 3.0–6.6), respectively for men with an increasing number of ISPs (0, 1, 2 and ≥ 3 ISPs)	Good	Yes
***G. Engström et al.*** [19]***; 2004***	Cohort, men; 46.8 ± 3.7.	6075	19	Nonfatal MI or death from CHD	A higher number of CHD deaths was noted in men who had presented a low-grade inflammation during many years before. Of the 680 men with a coronary event, 197 died the first day and 228 died within 28 days. The proportions who died the first day were 26%, 25%, 29%, and 35%, respectively, for men with an increasing number of ISPs (0, 1, 2 and ≥ 3 ISPs). The corresponding proportions who died within 28 days were 30%, 31%, 34%, and 38%, respectively	Good	Yes
***Verma et al.*** [20]***; 2005***	Cohort, men and women; 50–59	250		Severity of coronary artery disease (CAD) and modifiable CAD risk factors	Verma et al. explored how serum levels of three antioxidants (vitamin C, bilirubin and CP) were related to CHD risk factors. A 7–18% decrease was observed in CHD patients with severe disease with increasing serum levels of the three antioxidants. In the same line, a decrease of 14–20% was objectified in triple vessel disease and of 39% in MI occurred with increasing serum CP in CHD patients, compared to the non-MI group. An inverse relationship was found between the three antioxidants studied and coronary risk factor suggesting that greater care in traditional risk factors would maintain a high level of these antioxidants	Poor	No
***Brunetti et al.*** [21]***; 2008***	Cohort, men and women; 65.8 ± 11.25	123		Left ventricular systolic function during the early phase of acute MI	Systolic dysfunction in ST elevation acute MI patients seems to be associated with an inflammatory response featured by a rise in plasmatic concentration of acute-phase proteins (APPs); increase in APPs concentrations seems to own a short-term prognostic relevance. CP values were the most significant markers of acute heart failure when compared with patients without systolic dysfunction (40.1 ± 9.7 vs. 31.4 ± 7.6 mg/dL, p < 0.001).	Fair	Yes
***Göçmen et al.*** [22]***; 2008***	Case-control, men and women; 56.31 ± 2.74 (men); 54.23 ± 1.55 (women)	26/26		CAD	High CP and low albumin levels were found to be independent risk factors for CAD.	Poor	Yes
***Kaur et al.*** [23]***; 2008***	Case-control, men and women; 41–60	50/30		CAD	Increase in the levels of CP in patients of CAD (Mean ±SD, 48.93 ± 4.44 mg/dl as compared to controls (32.25 ± 4.67 mg %). CP could be a risk factor of CAD by modifying of Low-density lipoprotein (LDL) to an atherogenic form.	Poor	Yes
***Deepa et al.*** [24]***; 2009***	Case-control, men; 43 (mean approximately)	100/50		Acute MI with Diabetes Mellitus (DM) and non-DM	CP levels were significantly higher in diabetic and non-diabetic MI patients as compared with controls (p < 0.001) suggesting that CP may act as an oxidative stress indicator.	Poor	Yes
***Kumar et al.*** [25]***; 2009***	Case-control, men and women; 61.8 ± 3.8 (cases); 60.5 ± 3.4 (controls)	165/165		MI	CP levels were higher in MI patients than controls.	Fair	Yes
***Tang et al.*** [26]***; 2010***	Cohort, men and women; 63± 11 approximately	3828	3.0	Subclinical myocardial necrosis	The presence of subclinical myocardial necrosis was associated with elevations in CP levels.	Good	Yes
***Tang et al.*** [27]***; 2012***	Cohort, men and women; 63± 11 approximately	4177	3.0	Incident major adverse cardiovascular events (MACE = death, MI, stroke) in stable cardiac patients.	Serum CP level was associated with higher risk of MI with a HR of 2.35, (95% CI 1.79–3.09) comparing the top quartile versus the lowest. CP remained independently predictive of MACE (HR 1.55, 95% CI 1.10–2.17). Genetic variants at the CP locus were not associated with prevalent or incident risk of CAD.	Good	Yes
***T. B. Grammer et al.*** [28]***; 2014***	Cohort, men and women; 62.5 ± 10 approximately	3253	4.0	Angiographic CAD and mortality from all causes and cardiovascular causes.	When the highest quartile for CP levels was compared to the lowest, HR for death from any cause was 2.63 (95% CI, 2.17–3.20), and HR for death from cardiovascular causes was 3.02 (95% CI, 2.36–3.86). The concentration of CP was therefore independently associated with increased risk of death from all and cardiovascular	Good	Yes
***Xue Bao et al.*** [29]***; 2018***	Cohort, men and women; mean 57 approximately	Sub-cohort 4658	17.7 ± 5.46	DM and CVD	CP levels, alpha1-antitrypsin and soluble urokinase plasminogen activator receptor predicted increased risk of CVD but not DM.	Good	Yes

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
