# Peer review of "Ceruloplasmin and Coronary Heart Disease—A Systematic Review"

_nutrients, 2020, doi:10.3390/nu12103219_

Round 1

Reviewer 1 Report

The authors of this manuscript performed systematic review to assess the association between ceruloplasmin and cardiovascular disease with a special emphasis on the association between circulating ceruloplasmin and coronary artery disease. Overall 18 cohort and case-control studies were identified and included in the analysis. In brief, the majority of included studies support an association between circulating levels of ceruloplasmin and the risk for cardiovascular disease. However, the authors seem to take the results with reservation in the light of observational nature of studies and lack of mechanistic studies explaining the observed association.

This reviewer totally agrees with the authors on the need to assess the association between ceruloplasmin and the risk for cardiovascular disease considering a potential role of elevated levels of this molecule in inflammation and increased oxidative stress – factors that predispose to cardiovascular disease. Furthermore, evidence available on the association between ceruloplasmin and cardiovascular disease is limited and there is a paucity of recent studies assessing the association between ceruloplasmin and cardiovascular disease. In this regard, this systematic review is in due time. I have several concerns that may be addressed by the authors:

  1. My main concern has to do with the fact that although the authors identified and analyzed a large number of studies, no quantification of the magnitude of risk associated with a given cerulopalsmin level was provided. In other words, I suggest the authors to add a meta-analytic component to the review, i.e. to perform a systematic review and meta-analysis. This approach would quantify the overall risk associated with ceruloplasmin, assess heterogeneity across the studies and the meta-regression component of the analysis would provide information on the dose-response relationship in this association.
  2. The authors state to have conducted a search in electronic databases (PubMed and EMBASE) searching them for studies published after 1990. In principle, systematic reviews search all literature available with no source, time or language restrictions, because the aim is to analyze all scientific literature available.
  3. The abstract section is extremely scant in information. Although the journal allows for small abstracts (not exceeding the 200-word limit), the abstract should be re-written to contain the main message of the study.
  4. Table 2 should be improved, both visually and in terms of content. Its heading may contain: author/publication year, type of study, sample size (cases/control), ceruloplasmin cut-off, length of follow-up, outcomes of interest, level of adjustment, type of association (positive, negative, neutral or U(or J) shaped) and quality of study.
  5. The limitations of the study should be re-written and expanded. I could not understand them.
  6. Conclusions of the study are rather long. They should be succinct and informative.

Reviewer 2 Report

The systemic review proposed is very interesting and deserves publication and it could be further improved addressing the following:

Title:

As per the authors’ claim, their manuscript focused on exploring the evidence about ceruloplasmin (CP) and Coronary Heart Disease (CHD) after 1990. On this basis, the title should be more specifically referred to CHD.

Main text:

1.The links between the references in the text and Table 2 are unclear. For example in Table 2, the studies described are not quoted with the reference citation (number), but the studies are cited only in the main text and quoted in the Reference list (numbers of citation), and this makes the reading of the manuscript not fluid since it is necessary to search for the number of the reference to which the quoted article in the text refers. Furthermore, there are some articles cited in the main text that are not quoted in the reference list. This the case of Xue Bao, Yan Borné, Linda Johnson, Iram Faqir Muhammad, Margaretha Persson, Kaijun Niu & Gunnar Engström. Comparing the inflammatory profiles for incidence of diabetes mellitus and cardiovascular diseases: a prospective study exploring the ‘common soil’ hypothesis. Cardiovascular Diabetology volume 17, Article number: 87 (2018) that is quoted in the text but not in the reference list.

Moreover, in the case of Engström et al, who have more than one article cited, the lack of the reference number of quoted citation articles makes the reading confusing.

2.DM is never fully written, but is only reported as DM (diabetes Mellitus), which is confusing. In addition, the Coronary heart disease is sometimes reported as CHD other times not.

Table 2

3.Why the study by Dadu et al 2013 (Ceruloplasmin and Heart Failure in the Atherosclerosis Risk in Communities (ARIC) Study Razvan T. Dadu, et al Circ Heart Fail. 2013 September 1; 6(5): 936) is not included in the systemic review? It appears valid, but it is not included in Table 2. Citing (add ref) which studies are not included in the Table 2 would make the article clearer.

4.the study by Enbergs et al.; 1998 has “yes” in Table 2 (column “Supports a direct relationship between CP and CHD”) but in the text, it is reported “no relation, when attempting to relate CP with the extent of atherosclerosis in coronarography”.

Discussion

The study of Li et al., 2018 (Li et al, The loss of copper is associated with the increase in copper metabolism MURR domain 1 in ischemic hearts of mice; Experimental Biology and Medicine 2018) suggests that during the pathogenesis of myocardial infarction (MI), “...COMMD1 would play a critical role in exporting Cu from the ischemic myocardium to the blood...”. In the blood, the increase of copper is associated with an increase in ceruloplasmin. This mechanistic study can be quoted in the manuscript to discuss the role of copper and ceruloplasmin in the pathogenesis of MI. The increase of copper (ceruloplasmin) in the blood caused by MI seems not (or at least not completely) associated with an inflammatory response.

Conclusion

Since apparently 1 (or 2) of 18 studies do not show evidence about the relation of ceruloplasmin to CHD, I think that the use of the term 'majority' is an understatement. Especially assuming that a study (Verma et al.; 2005) seems to have some weaknesses. Perhaps it could be said that there is a consistency in the results, although one study is not in line. Maybe the authors have drawn their conclusions in line with systemic review guidelines, in this case, it would be better to specify. Furthermore, a meta-analysis study would certainly be useful to improve knowledge in this regard. Animal studies investigating the mechanisms of CP involvement in CHD can help figure out the biological plausibility, as in the case of Li et al 2018.

Round 2

Reviewer 1 Report

No further comments.

Reviewer 2 Report

The revised manuscript is improved